# Maritime Over the Horizon Sensor Integration: HFSWR Data Fusion Algorithm

**Dejan Nikolic [1,*]**[iD]**, Nikola Stojkovic [2], Zdravko Popovic [1], Nikola Tosic [3], Nikola Lekic [1], Zoran Stankovic [4] and Nebojsa Doncov [4]**

1    Vlatacom Institute; Belgrade 59714, Serbia; zdravko.popovic@vlatacom.com (Z.P.); nikola.lekic@vlatacom.com (N.L.)

2    School of Electrical Engineering, University of Belgrade, Serbia and Vlatacom Institute, Beograd 101801, Serbia; nikola.stojkovic@vlatacom.com

3    Faculty of Transport and Traffic Engineering, University of Belgrade, Serbia and Vlatacom Institute, Beograd 101801, Serbia; nikola.tosic@vlatacom.com

4    Faculty of Electronic Engineering, University of Nis, Nis 700132, Serbia; Zoran.Stankovic@elfak.ni.ac.rs (Z.S.); nebojsa.doncov@elfak.ni.ac.rs (N.D.)

\*    Correspondence: dejan.nikolic@vlatacom.com; Tel.: +381-11-377-1100

**Abstract:** In order to provide a constant and complete operational picture of the maritime situation in the Exclusive Economic Zone (EEZ) at over the horizon (OTH) distances, a network of high frequency surface-wave-radars (HFSWR) slowly becomes a necessity. Since each HFSWR in the network tracks all the targets it detects independently of other radars in the network, there will be situations where multiple tracks are formed for a single vessel. The algorithm proposed in this paper utilizes radar tracks obtained from individual HFSWRs which are already processed by the multi-target tracking algorithm at the single radar level, and fuses them into a unique data stream. In this way, the data obtained from multiple HFSWRs originating from the very same target are weighted and combined into a single track. Moreover, the weighting approach significantly reduces inaccuracy. The algorithm is designed, implemented, and tested in a real working environment. The testing environment is located in the Gulf of Guinea and includes a network of two HFSWRs. In order to validate the algorithm outputs, the position of the vessels was calculated by the algorithm and compared with the positions obtained from several coastal sites, with LAIS receivers and SAIS data provided by a SAIS provider.

**Keywords:** Radar; HFSW radar; OTH radar; radar tracking; data fusion; AIS; marine systems

---

## 1. Introduction

In recent years, organized crime in the maritime arena committed away from territorial waters practically flourished, threatening both the secure flow of goods from the Exclusive Economic Zones [1] (EEZ) and also the lives of the participants in marine traffic. All marine nations are obligated to fully control the entirety of their respective EEZ, and not only their territorial waters. Moreover, in some areas of the world, the situation is so serious that it requires UN [2] and/or EU intervention [3], since nations which have jurisdiction over those waters are helpless. Since EEZs are huge bodies of water which can cover hundreds of thousands of square miles, complete monitoring is much easier said than done.

To the best of our knowledge, there are only two ways to achieve complete EEZ monitoring. The first approach utilizes optical and microwave sensors on platforms such as satellites and airplanes, thus avoiding the limitations of the sensors, but this introduces limitations in the platform. The most

limiting factor is the interrupted data availability, since no airplane is able to stay in the air constantly during the whole year and during all weather conditions. Meanwhile, satellites, which are orbiting around the earth, will be over the zone of interest for a limited time only. The other approach uses a network of HFSWRs [4,5] to ensure the constant surveillance well beyond the horizon. Since the price of the HFSWR radar network is significantly lower than the combined cost of the aforementioned sensors, and because of the fact that those sensors provide limited data availability throughout the whole year, it is clear why HFSWR networks are slowly becoming the sensors of choice for maritime surveillance at OTH distances.

The data obtained from the HFSWRs must be processed before they can be combined. The tracking algorithms at the single radar level are used for that purpose. There are several types of tracking algorithms, which are approved for radars [6–8]. Moreover, a radar network for over-the-horizon sea surveillance utilizes HF radars and introduces additional challenges to the tracker design [4,5,9–13]. A single radar tracking algorithm used to pre-process data for the algorithm presented in this paper is described in [13].

When individual radar targets are ready for multi radar data fusion there are several algorithms which can be used for multi-radar multi-target data fusion [14–18]. The algorithm described in this paper is based on an algorithm presented in [18] and represents its successor. Based on a priori tactical knowledge (the maneuverability of the vessels of interest), a simple yet effective way for the data fusion process is chosen. This data fusion process relies on the weighted minimum mean square error (MMSE) for vessel position calculations and the Thiel–San estimator for the tracking process. While the correctness of the algorithm described in [18] is demonstrated on simulated data, the correctness of the algorithm described here is demonstrated with the data obtained from the sensors working in a real operational environment, precisely the Gulf of Guinea. The used data set is collected from all the available sensors in the operational region during a one-year period. Moreover, in order to show the validity of the vessels' positions calculated by the algorithm, AIS [19,20] data is used. AIS reliability and accuracy is not questioned in developed countries, since in those waters the behavior of the participants in marine traffic is well regulated and controlled by governmental bodies. In the Gulf of Guinea this is not the case. This paper will not deeply examine irregular or corrupted AIS data and the integration of such data into the maritime surveillance system, since this has already been done in [21]. Here, we will use data which is confirmed to be reliable in order to analyze some interesting cases during the HFSWR data fusion process. It is worth noting that the algorithm described in this paper already passed the final stage of operational testing and has been in everyday operational use for over a year.

The rest of the paper is organized as follows: in Section 2, we briefly describe HFSWR resolution capabilities as well as the operational environment. In Section 3, we give a description of the data fusion algorithm. The track modeling process is given in Section 4. Field results are presented in Section 5, statistical analyses are given in Section 6 and we draw conclusions in Section 7.

## 2. Operational Environment and HFSWR Resolution Capabilities

The environment where the whole system is deployed is the Gulf of Guinea. From our point of view, this environment is one of the most challenging in the world for the task we are targeting. A detailed environment description may be found in [21], while a full description of the HFSWR that is used may be found in [22].

It is important to point out the resolution and coverage of HFSWRs which are used:

- The range resolution is 1.5 km,
- The angle resolution varies from $0.3°$ in the center of the coverage area (the line that is practically perpendicular to the receive array) up to $2.5°$ at the edges of the coverage area (the areas which are practically located at $\pm 60°$ from the line perpendicular to the receive array).
- The nominal range is 80 nautical miles (approx. 150 km) for the Bonn express class of vessel [23] during night-time and sea states [24] up to 3. For larger vessels and during the day-time, the range

can extend even beyond 125 nautical miles (approx. 230 km). The angle coverage is set to 120 degrees, regardless of the vessel's size or the time of day.

- The network coverage area is currently the western part of the Gulf of Guinea, and it is shown in Figure 1.

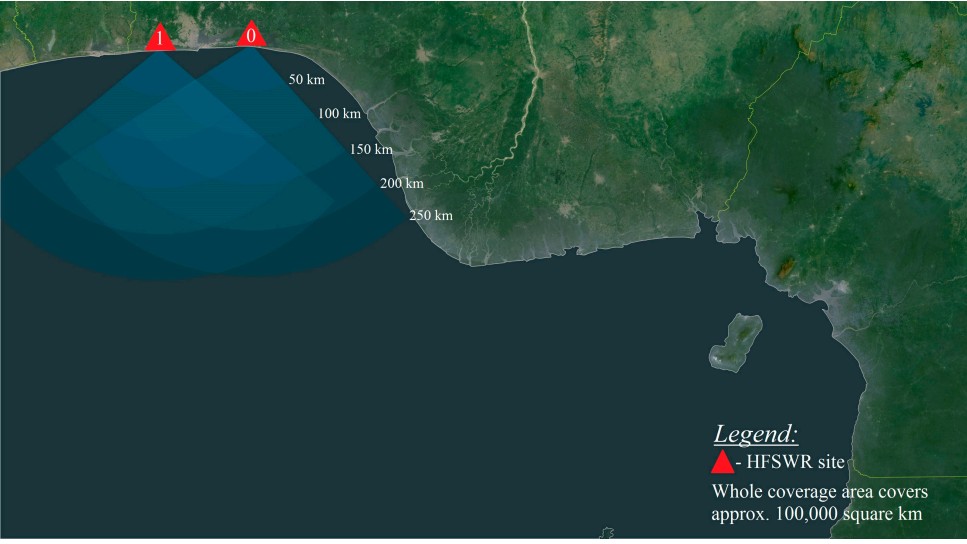

**Figure 1.** HFSWR network coverage area (taken from [21]).

Based on the aforementioned HFSWR resolution capabilities, it becomes clear that the positions of newly formed targets (detected vessels) may differ significantly. For example, let's say that a vessel is detected for the first time by both radars approx. 200 km offshore at an azimuth angle of $40°$. Since the angle resolution at $40°$ is about $1.3°$, the arc length at that distance is nearly 5 km. Since the range resolution is 1.5 km, it means that the resolution cell covers an area of about 7.5 km$^2$. The same is valid for the second radar as well. Since the resolution cells of the two radars are mutually overlapping, the reported positions may range from 7.5 km$^2$ up to nearly 15 km$^2$. This makes the fusion of such vastly separated targets quite a challenging task. It is also important to note that vessel positions for long and stable tracks (vessels which are tracked by a certain radar for a few hours) are quite precise and do not differ more than 500 m from the actual vessel position (the one reported by AIS).

In the end, in order to verify the vessel's position and thus demonstrate the proposed algorithm accuracy, the data obtained via the following sensors will be used:

- Land AIS data—provided by six coastal sites equipped with AIS receivers and
- Satellite AIS data—provided from the SAIS provider Orbcomm [25].

Please note that the vessel's positions are represented with geographical longitude and latitude.

## 3. Description of Data Fusion Algorithm

Each radar in the radar network performs its own tracking procedure, as described in [13]. The tracking procedure relies on Joint Probabilistic Data Association–Unscented Kalman Filtering (JPDA–UKF) algorithms. The HFSWR data-fusion algorithm is an iterative process triggered when a set of single radar tracks from any radar in the network is received.

Each track formed by any radar in the network has an identification (Id) assigned by the radar which formed the track. When the track appears for the first time, it is placed in the so–called track-waiting list. All tracks in the track-waiting list are not forwarded for further processing, until they appear N times in M consecutive periods. After a radar track passes this threshold, it is moved from the track-waiting list to the tracking list.

The data fusion algorithm creates so-called fused tracks and operates with them. A fused track is formed from at least one single radar track. Upon the formation of a fused track the algorithm assigns a new Id to the fused track. Besides this new Id, every fused track maintains the list of Ids that represent the identifications of radar tracks which are used to form it. It is important to note that one fused track may not contain more than one track originating from the single radar.

The main steps of the data fusion algorithm are shown in Figure 2.

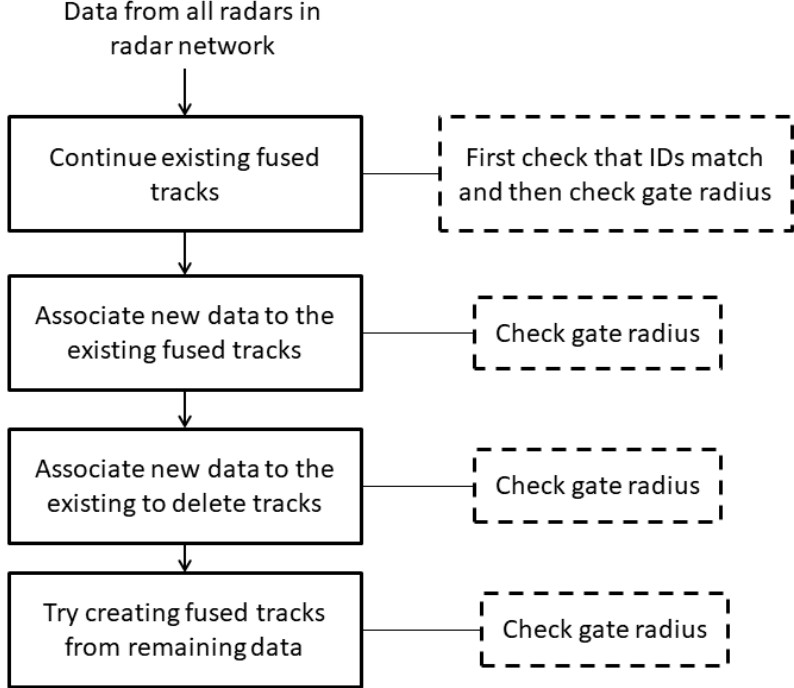

**Figure 2.** Main steps of the data fusion algorithm (taken from [18]).

During the first step, the algorithm is trying to update already formed fused tracks using newly arrived radar tracks. For each fused track algorithm, it is checking if the originating radar tracks are present in the newly arrived data. If the Id is matched, the algorithm checks if the radar track still falls within the fused track gate radius. This gate radius is calculated based on the vessel's speed, so only the range radius is taken into account. From our point of view, since all single radar tracks are already preprocessed by JPDA–UKF algorithms, there is no need to include the state vector again. When both conditions are satisfied, the fused track is updated. If at least one of the criteria is not satisfied, the radar track is forwarded to the following step of the algorithm.

During the second step of the algorithm, all previously unassigned radar tracks that are considered potentially suitable for an association with existing fused tracks are processed. This means that for all of the unassigned radar tracks, the gating with existing fused tracks is checked. Please note that one fused track may not have more than one radar track coming from each radar. For simplicity's sake, this procedure is shown in Figure 3.

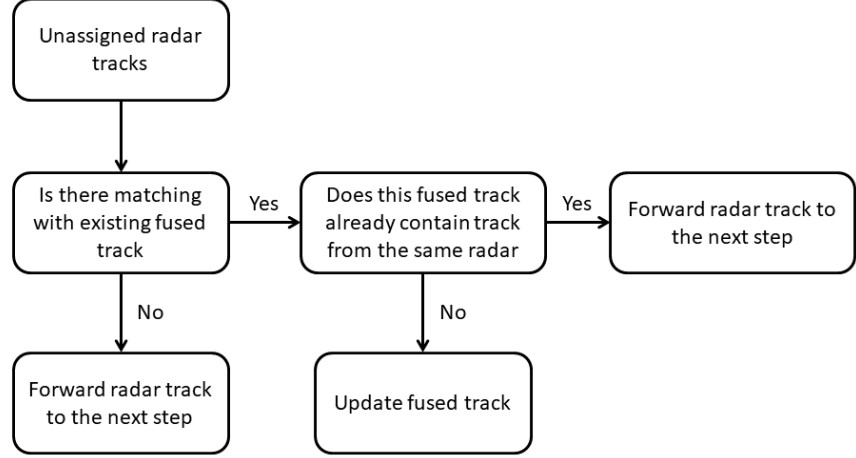

**Figure 3.** 2 of the data fusion algorithm.

If a fused tracks loses all of its source radar tracks, it is moved to the so-called predicted track list. The fused track stays in this list for K consecutive update periods, and it is referred to as the predicted track. If no source (either new or old) is found within this time frame, the track is declared non-existent and the algorithm stops further processing.

The third step of the algorithm operates on predicted tracks in a manner similar to the previous step of the algorithm: all the tracks which are currently in the predicted track list are compared with the remaining radar tracks. If a matching track is found, the track is returned to the fused track list.

During the last step of the algorithm, new fused tracks are formed from the remaining unassigned radar tracks. It is important to remember that the fused track may not have more than one track originating from each of the radars.

At the end, the predictions and corresponding gate radiuses for all of the tracks present in the fused track and predicted track lists are calculated. These predictions as well as the gate radiuses will be used when the next data set from radars arrives, in order to create, maintain or delete fused tracks.

## 4. Track Modeling Process

Since EEZ monitoring is our goal, vessels which are the most interesting to us are vessels used for the transportation of goods, such as various types of cargo vessels and tankers. The typical characteristics of these vessels are:

- Most are very large and their length is often more than 100 m, while their displacement is usually more than 50,000 DWTs.
- The top speed seldom exceeds 25 knots, while the usual cruising speed ranges from 10 to 20 knots (sometimes even less than that).
- Most of the time they are traveling along a straight line, and when they make turns they do so slowly and in a wide arc.

It is clear that the vessels of our interest are easy to track. On the other hand, before the vessels can be tracked at all, the radar tracks need to be fused into fusion tracks. Since the positions of two radar tracks originating from the same vessel may differ significantly, while the actual vessel's position is often somewhere between them, the fusion algorithm is based on the MMSE approach. Furthermore, in order to favor a radar track whose position best matches the actual vessel's position, the weighting factor is introduced into the fused track position calculations. This weighting factor is called the *confidence level (CL)*, and it represents the reliability of radar data and takes values ranging between 0 and 1. The confidence level for each radar track is calculated with the following equation:

$$CL = \frac{CL_{snr} + CL_{ang} + CL_c}{3} \tag{1}$$

where $CL_{SNR}$ represents the confidence level based on the signal to noise ratio of the target's detection and is calculated in the following manner:

$$CL_{SNR} = \begin{cases} \frac{(SNR-10)}{50}, & SNR < 60 \ dB \\ 1, & SNR > 60 \ dB \end{cases} \tag{2}$$

It is worth noting that target will not be detected by a radar if its SNR is below 10 dB, while all of the targets with an *SNR* greater than 60 dB are considered stable. $CL_{ang}$ represents the confidence level based on the relative angle between the target and the radar, and its values are stored in a table. $CL_c$ represents the confidence level based on the consistency of the radar tracking, and it is calculated in the following manner:

$$CL_C = 0.05 + \begin{cases} 0.05, & \text{until it reaches 1, if the track data are based on detection} \\ -0.1, & \text{until it reaches 0, if the track data are based on prediction} \end{cases} \tag{3}$$

At the end, the fused track (vessel) parameters (geographical latitude and longitude) are calculated as follows:

$$Lat_o = \frac{\sum_{i=1}^{n} CL_i \cdot Lat_o(i)}{\sum_{j=1}^{n} CL_j}, \tag{4}$$

$$Long_o = \frac{\sum_{i=1}^{n} CL_i \cdot Long_o(i)}{\sum_{j=1}^{n} CL_j}, \tag{5}$$

where $CL_i$, $Lat_o(i)$ and $Long_o(i)$ are the confidence level, latitude and longitude, respectively, reported by the *ith* radar; and *n* is the number of radars detecting the target, i.e., the total number of radar tracks that are needed to be fused into a single fused track.

With the radar tracks fused into fused tracks, predictions for their next occurrences need to be made. Since the real vessel is traveling in a straight line most of the time, while radar detections and thus radar tracks "fall" around that line, we decided to keep the tracking process as simple as possible and rely on the algorithm based on a linear regression [26]. In particular, due to its robustness and good tolerance to outliers, the Thiel–Sen Estimator [27,28] is used during the prediction calculations, in the following manner.

When the first pair of the data set ($Lat_o(i)$, $Long_o(i)$) is available for a given fusion track estimator, it tries to determine the median *m* of the slopes using Equation 4:

$$m = \frac{Long_o(i) - Long_o(i-1)}{Lat_o(i) - Lat_o(i-1)} \tag{6}$$

With *m* determined, the *b coefficient* may be determined as the median of the following values:

$$b = Long_o(i) - mLat_o(i) \tag{7}$$

The projected vessel's course is calculated as:

$$course = arctan(m) \tag{8}$$

The estimator will then enlarge the data set with each available pair of $Lat_o(i)$ and $Long_o(i)$ and repeat the procedure. It is important to note that the data set will not be enlarged indefinitely, but only until a certain number of data pairs (T) is reached and the so–called data window is formed. The reception of any new data pairs after the threshold is reached will cause the erasure of the oldest data pair, thus sliding the window forward. In this way, if the vessel starts any maneuver, the old data points will have less influence in the tracking process. Next, the true vessel's velocity (*Vel*) needs

to be calculated. This is done by dividing the distance the vessel covered by the elapsed time, in the following manner:

$$V_{el} = \frac{\sqrt{(K_1(Lat_o(i) - Lat_o(1))^2 + (K_2(Long_o(i) - Long_o(1))^2}}{t} \qquad (9)$$

where $Lat_o(i)$ and $Long_o (i)$ represent the vessel's parameters at the current time, $Lat_o(1)$ and $Long_o (1)$ represent the initial vessel's parameters in the current window (not initial in the entire track) and t represents the elapsed time. The coefficients $K_1$ and $K_2$ are calculated according to [29].

In order to predict the new vessel's position, the current location on the slope is moved for a distance which the vessel is able to cover in one detection cycle. In this case, the detection cycle is 33 s. At the end, the gating radius around the predicted vessel's position is a circle of a radius equal to half of the covered distance.

At the end of this section, the computational complexity of this algorithm needs to be addressed.

There are two factors that influence the computational complexity of the algorithm: N (number of active tracks per radar) and M (number of radars with overlapping zones). The functions that form the algorithm have different complexities ranging from $O(N)$, $O(N \times M)$, $O(N^2)$, $O(N^2 \times M)$ all the way to the most complex with the complexity: $O(N^2 \times M^2)$. Since the number of radars that have overlapping zones shouldn't be more than 3, we can consider that M << N, and that M and $M^2$ are constant factors in the complexity analysis. Therefore, the complexity of the algorithm can be considered to be: $O(N^2)$. One additional thing to note here is that this complexity is valid only during the initial phase of the algorithms' operation. One N in $N^2$ factor is for old/stable tracks, while the other represents new tracks which arrived during the last period of integration. The number of newly appeared tracks in the stable mode of operation is usually at least an order of magnitude lower than the number of old tracks. Therefore, we may assume that the overall algorithm complexity in the stable mode of operation is close to $O(N \times \log(N))$.

## 5. Field Results

Before the field results are presented, a few things need to be noted:

1.  The graphical environment presented here is used just for the data visualization of the described sensor fusion process, not as the command and control software's GUI.
2.  The graphical elements are the following:

    a.  The yellow markers always represent radar tracks, both fused and single radar.
    b.  The white markers always represent AIS data.
    c.  The hexagonal encirclement around the marker means that that marker is selected, and details about the vessel are shown in a separate window.
    d.  The trace colors are randomly determined and don't have any particular significance.

3.  Please note that each vessel has a unique Maritime Mobile Service Identity (MMSI) number assigned to it [30].

In order to demonstrate the algorithm's capabilities in a real working environment, a few possible scenarios will be described here. These scenarios are:

1.  "Clear case"—Both radars provide stable and long living tracks.
2.  "New update"—One radar provides a stable and long living track while the second radar just detected a new target which falls into the gating radius.
3.  "Separated tracks"—Both radars track a target, but the positions they are providing are significantly different.
4.  "Unstable target"—The radar(s) are tracking a target, but the data is erratic.

First, a so called "clear case" will be presented. In such cases both radars are providing stable tracks with minimal position differences (Figure 4).

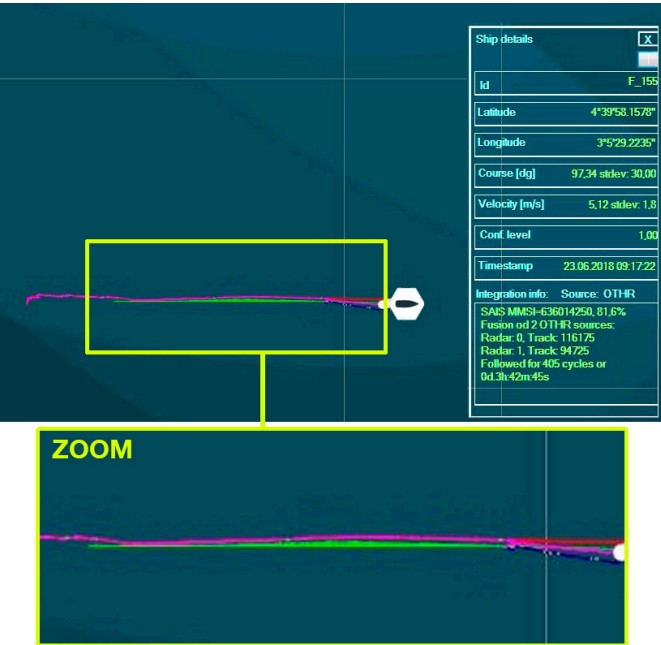

**Figure 4.** Case" (dark blue trace: radar 0 data, green trace: radar 1 data, purple trace: fused track, red trace: AIS data).

Fused track id No. F_155 consists of radar 0 track id No. 0_116175 and radar 1 track id No. 1_94725. Both radars are providing stable and long-lasting tracks with minimal position differences so the data fusion process is quite simplified. Moreover, there is only one AIS target nearby (MMSI 636014250) which further justifies the data fusion decision. The confidence level is listed in the fused track data window, representing the highest confidence level among all single radar track confidence levels.

Next, a new update of an existing track will be discussed. This situation occurs when only one radar provides the data for the fused track for a long time, and a radar track from another radar appears within the fused track gating radius (Figure 5).

As can be seen from Figure 5, the fused track id No. F_1240 is regularly updated by radar 1 id No. 1_1697622, so the traces are nearly overlapping. Upon the arrival of a new radar track id No. 0_1042366, originating from radar 0, the fused track shifts slightly towards the new track. However, it is still more aligned to the long track, since it has a longer tracking history and thus a higher confidence level. It is important to note that the long and new tracks are originating from different radars, as otherwise they will not be considered for the fusion process. Furthermore, a fusion of two significantly separated radar tracks is presented. In the situation shown in Figure 6, both radars are tracking the same target; however, the vessel positions they are providing are mutually separated by approximately 3 km.

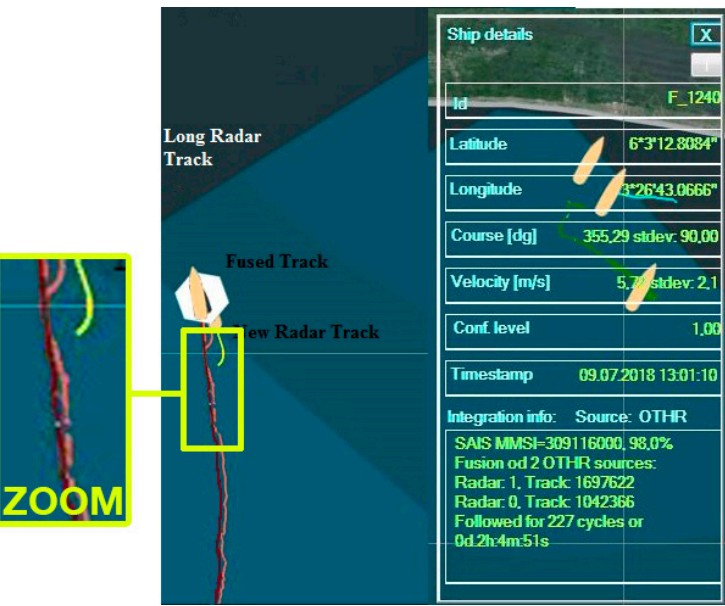

**Figure 5.** Update" (red trace: long radar track provided by radar 1, orange trace: fused track, yellow trace: new radar track provided by radar 0).

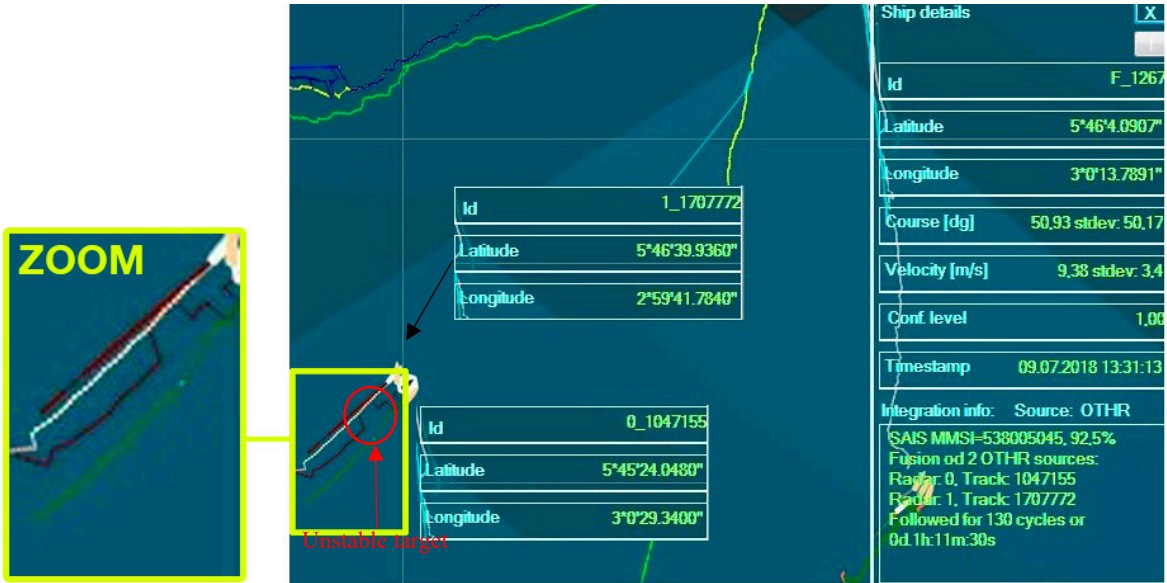

**Figure 6.** "Tracks" and "Unstable target" (red trace: radar 0 data, green trace: radar 1 data, brown trace: fused track, white trace: AIS data).

From Figure 6, one can see that the fused track id No. F_1267 is formed from radar 0 track id No. 0_1047155 and radar 1 track id No. 1_1707772, despite the radar tracks' mutual distance. This example also demonstrates an "unstable target" situation very well, since the fused track is maintained in spite of radar 0's erratic data availability. It can be seen that radar 0 stopped providing target data for some time, as highlighted by the red circle. During that period, the fused track was maintained entirely by the data available from radar 1; hence, the fused track shifted toward the radar 1 track. Upon the reappearance of the radar 0 data, both radar tracks were reintegrated into a single track. This decision is justified since there is only one AIS target in the vicinity (MMSI 538005045). In this way, multiple false (duplicated) targets are eliminated, even though with each reappearance of the radar 0 target a new radar id was assigned.

In the end, it is important to emphasize that the presence or absence of AIS data does not influence the fusion process. The AIS data presented here are used only to justify the decisions made during the fusion process.

## 6. Statistical Analyses

In this chapter, the statistical analysis of a number of targets received from the HFSWRs in the network and a number of targets forwarded to the further processing is presented. In other words, this analysis shows the percentage of duplicate targets eliminated by the presented algorithm.

Before presenting the analysis, a few things need to be pointed out:

- The period of a whole year is analyzed here. Starting with 1st of January 2018 and ending with 31st of December 2018.
- Power supply issues are common in the Gulf of Guinea—these are not isolated incidents. Despite the fact that all sites are equipped with UPSs which can power the equipment for approximately 24 h, some sites are located in remote areas and cannot be reached within 24 h. This leads to situations where a significant drop of the number of detected targets from one site is present, simply because the site was with no power.
- A similar situation occurs when there is a major problem with the satellite links. The problem mostly occurs when storms are raging in a certain area blocking the satellite communications from sites located in that area. This also causes a significant drop of targets as long as the link is down.
- Taking the above two points into account, it can be expected that the number of targets dwindles from time to time and thus creates some sharp differences between adjacent months.

In Figure 7, one log file created during the data fusion process is presented.

```
***********************************************************************************************
Iteration: 2291; Time: 2018-07-17 21:11:13; # FOTHR (total): 17; # FOTHR with AIS: 11; # RAD0: 15; RAD1: 7;
***********************************************************************************************
Total situation on all ever appeared targets:
Total FOTHR=151; integrated=92
Total RAD0=109; Total RAD1=83;
***********************************************************************************************
```

**Figure 7.** One iteration log file.

Explanation of the log file fields:

- Iteration –number of iteration processes during a single day
- Time—time and date when the log file was created
- FOTHR (total)—number of fused tracks in the current iteration
- FOTHR with AIS—number of fused tracks which have corresponding AIS data in the current iteration
- RAD 0—number of tracks received from radar 0 in the current iteration
- RAD 1—number of tracks received from radar 1 in the current iteration
- Total FOTHR—number of fused tracks since the beginning of the day
- Total RAD 0—number of tracks received from radar 0 since the beginning of the day
- Total RAD 1—number of tracks received from radar 1 since the beginning of the day
- Integrated—number of fused tracks which have corresponding AIS data since the beginning of the day

Based on the data displayed in Figure 7, the following can be concluded:

1. The total number of targets received from all HFSWRs during this iteration is 22,
2. The total number of fused targets is 17,
3. The number of eliminated targets is 5.

This means that 23% of all targets were duplicated and could cause false alarm triggers in command and control (C2) systems.

The log files are collected during the day in order to present the statistics for that day. One daily log file is presented in Figure 8.

```
************************************************Daily report***************************************************
Daily report - Iteration: 2487; Time: 2018-06-27 00:00:12; # FOTHR: 173; # FOTHR with AIS: 80; # RAD0: 137; RAD1: 74;
************************************************************************************************************************
```

**Figure 8.** Daily log file.

For this day, RAD 0 detected 137 targets, while RAD 1 detected 74 targets, so the HFSWR network detected 211 targets. Overall, 173 targets were delivered to the C2 system, and 80 of those targets have corresponding AIS data, while 38 radar targets were eliminated as duplicates. This means that nearly 20% of all received targets were duplicated. Please note that in this particular case there were more integrated HFSWR tracks than AIS data. The situations vary from day to day.

The day log files are collected on a monthly basis starting from the 1st of January 2018 and ending with the 31st of December 2018, in order to create a statistical analysis, which is presented in Figure 9.

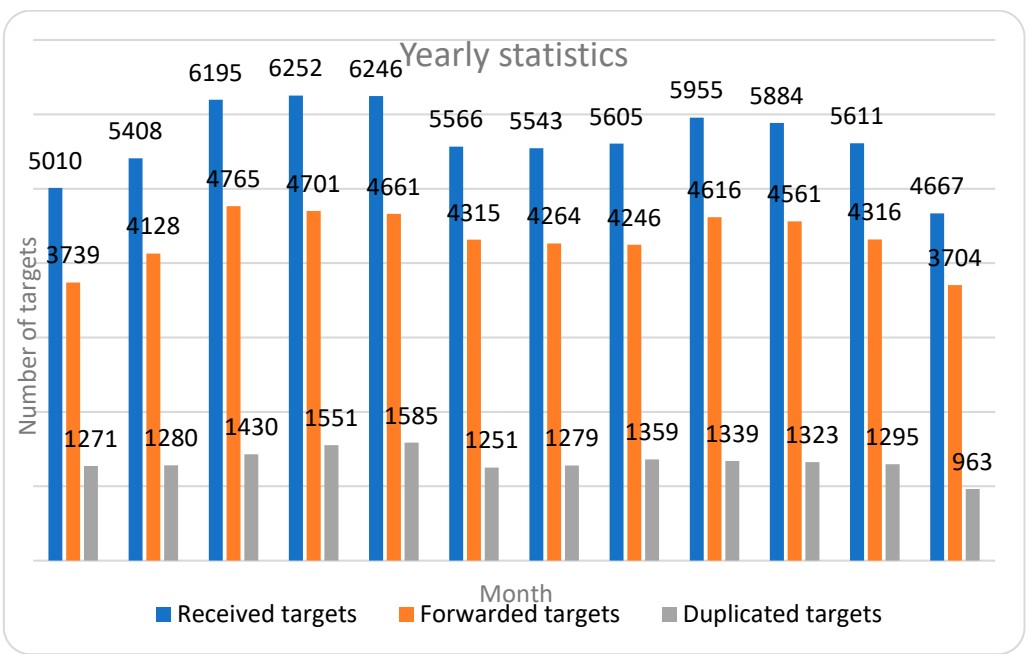

**Figure 9.** Yearly statistics.

Figure 9 shows that around 1200 to 1600 duplicate targets were eliminated every month. Some drops are particularly noticeable during December and January, which can be attributed to the holiday season. In total, for the observed period, the proposed algorithm received 67,799 single radar tracks and, after processing, 52,016 fused tracks were delivered to the C2 system. This means that 15,783 radar tracks were eliminated, since they were duplicates. Overall, it can be concluded that the presented algorithm eliminates approximately 23.3% of the received data, thus significantly reducing the false alarm rate.

## 7. Conclusions

In this paper we presented, described and tested an algorithm for HFSWR data fusion at OTH distances. The testing environment is located in the Gulf of Guinea and includes a network of HFSWRs consisting of two HFSWRs, several coastal sites with LAIS receivers and SAIS data provided by a SAIS data provider. The proposed algorithm is based on MMSE principles and once the data is fused,

the algorithm relies on the Thiel–San estimator in order to successfully track targets. Besides this rather simple approach, the algorithm relies on a priori tactical knowledge regarding the target model and its maneuverability in order to solve the situation in the field. After comprehensive tests, which were performed for over a year, it may be said that the proposed algorithm shows a high reliability and efficiency in spite of its simplicity. This approach yields more accurate tracking, prevents the creation of duplicate targets and delivers unique a radar picture at OTH distances at very low computational costs.

For future work, we intend to examine the possibility of algorithms based on neural networks in the HFSWR data fusion process.

**Author Contributions:** Conceptualization, D.N.; Data curation, N.S.; Formal analysis, D.N.; Investigation, D.N., Z.S. and N.D.; Methodology, D.N.; Project administration, N.L.; Resources, N.T.; Software, N.S. and Z.P.; Validation, N.L.; Visualization, N.S. and Z.P.; Writing—original draft, D.N.; Writing—review & editing, D.N., Z.P. and N.T.

**Funding:** This research was funded by Vlatacom Institute. The APC was funded by Vlatacom Institute.

**Acknowledgments:** Research is completely funded by Institute Vlatacom. Furthermore, APC is also funded by Institute Vlatacom.

**Conflicts of Interest:** The authors declare no conflict of interest.

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
