# Peer review of "Maritime Over the Horizon Sensor Integration: HFSWR Data Fusion Algorithm"

_remotesensing, doi:10.3390/rs11070852_

Round 1

Reviewer 1 Report

The strong side of the paper is relying on real data recorded form existing system in which the method is implemented. Congratulations on that. In general the work is done and the research are provided properly. In my opinion the literature review could be deepened and some issues in the method itself should be clarified.

Detailed remarks:

- line 52 - there are many various algorithms for radar tracking, which are nto included. For example use of neural networks for this purpose could be mentioned.

- line 57 - other interesting approach can be find in [3] Stateczny A., Kazimierski W., (2011), Multisensor tracking of marine targets – decentralized fusion of Kalman and neural filters, International Journal of Electronics and Telecommunications, vol. 57, No 1, pp. 65 –

- line 62 - the reliability of AIS itself should be commented, if it is used as reference

- line 123/127 - gating is based only on range radius. Usually not only position is used for association but also state vector and others. PLease comment on it. Why only this basiv, very simple apporach is used. IS it the specifity of HFSWR?

- line 165 - detailed explanation of how the coeficients are calculated should be given - line 168 - how is the consistnecy of radar measured/calculated?

- line 178 - linear regression can be used as basic in this case, however othoer, more sofisticated model should be included in future

- line 187 - this equation is not obvious. usually the approach based on arctan(deltalambda/deltafi) is used. Please rpovide explanation of this equation (likely with teh drawing)

- line 195 - what is the unit of this measruement? [degrees/seconds]? sounds strange

Author Response

Dear reviewer,

Please find answers to your questions below.

Best regards,

Dejan Nikolic

The strong side of the paper is relying on real data recorded form existing system in which the method is implemented. Congratulations on that. In general the work is done and the research are provided properly. In my opinion the literature review could be deepened and some issues in the method itself should be clarified.

Thank you for this very positive review. We have accepted all your remarks and extended the paper and references accordingly. Please refer to the comments below for details.

Detailed remarks:

- line 52 - there are many various algorithms for radar tracking, which are nto included. For example use of neural networks for this purpose could be mentioned.

Additional references are added.

- line 57 - other interesting approach can be find in [3] Stateczny A., Kazimierski W., (2011), Multisensor tracking of marine targets – decentralized fusion of Kalman and neural filters, International Journal of Electronics and Telecommunications, vol. 57, No 1, pp. 65 –

Additional references are added.

- line 62 - the reliability of AIS itself should be commented, if it is used as reference

Accepted.

Some comments regarding quality of AIS service are added in lines 72/79. Moreover, reference describing it in more details are added.

- line 123/127 - gating is based only on range radius. Usually not only position is used for association but also state vector and others. PLease comment on it. Why only this basiv, very simple apporach is used. IS it the specifity of HFSWR?

Thanks for the remark. We extended the paper with lines 143/145 in order to address this issue.

- line 165 - detailed explanation of how the coeficients are calculated should be given - line 168 - how is the consistnecy of radar measured/calculated?

Accepted.

Explanation is added.

- line 178 - linear regression can be used as basic in this case, however othoer, more sofisticated model should be included in future

Thanks for the comment, future work is added at the end of the conclusion (lines 390/391).

- line 187 - this equation is not obvious. usually the approach based on arctan(deltalambda/deltafi) is used. Please rpovide explanation of this equation (likely with teh drawing)

Thank you for this remark. Wrong equation is written in the first paper. Correct approach is based on arctan. This is now corrected.

- line 195 - what is the unit of this measruement? [degrees/seconds]? sounds strange

Again, thank you for pointing this out. There was typo in the equation, since coefficients K1 and K2 were omitted. These coefficients are needed to determine Euclidian distance based on ellipsoidal Earth model. With this correction it becomes clear that speed is expressed in m/s.  

Reviewer 2 Report

Reviewers' Suggestions and Comments:

The methodologies adopted for the problems have some novelties. However, there are some deficiencies in the paper as the followings:

Q1. The motivation of how to use (or not use) the theoretic results in practice is not quite clear and should be further emphasized. The problems discussed in this paper have been studied by others. Hence the importance of the particular problems in this paper should be clearly addressed.

Q2.   For the results presented in the examples (Figures and/or Tables), more explanations on them seem necessary and would help readers to follow the development of the results.

Q3. It is desirable to analyze or comment the computational complexity. The paper does not contain any innovations. All the results are based on the simulations. There are no analytic evaluations.

Q4. A better clarification of paper's novelty-- what exactly new being proposed in this paper and what is being resolved.

Q5. The more effectiveness and efficiency of the results and techniques in this paper compared with other existing ones in literature should be further addressed.

For these reasons, I recommend Major Compulsory Revisions the submission.

Author Response

Dear reviewer,

Please find point to point response to your comments further in this e-mail.

Best regards,

Dejan Nikolic

Q1. The motivation of how to use (or not use) the theoretic results in practice is not quite clear and should be further emphasized. The problems discussed in this paper have been studied by others. Hence the importance of the particular problems in this paper should be clearly addressed.

Thank you for your remark.

According to your remarks we extended the paper with lines 62/71 in order to better present the used method.

The main motivation was to present a simple yet effective way to achieve HFSWR sensor fusion at the horizon distances.

The proposed method is justified with data collected during one year period from a real system located in the Gulf of Guinea.

At the end we added more references in order to show other solutions to the presented problem.

Q2.   For the results presented in the examples (Figures and/or Tables), more explanations on them seem necessary and would help readers to follow the development of the results.

Accepted.

Figures are edited in order to clarify details.

Q3. It is desirable to analyze or comment the computational complexity. The paper does not contain any innovations. All the results are based on the simulations. There are no analytic evaluations.

Accepted.

Comment regarding computational complexity is added in lines 213 to 225.

Thank you for this remark. We have more clearly explained that results presented in this paper are based on experiment using a real sensor network located in the Gulf of Guinea. This experiment lasted for one year.

Q4. A better clarification of paper's novelty-- what exactly new being proposed in this paper and what is being resolved.

Thank you.

According to your remarks we extended the paper with lines 62/71 in order to better present the used method.

The main motivation was to present a simple yet effective way to achieve HFSWR sensor fusion at the horizon distances.

In order to better clarify benefits of the proposed algorithm, the paper is extended with line 376/380.

Q5. The more effectiveness and efficiency of the results and techniques in this paper compared with other existing ones in literature should be further addressed.

System which is described in this paper is live and fully operational now. It is not possible to make any major changes, like implementation of new fusion methods.

However, for future work we plan to examine other methods and implement them in new systems. The paper is extended with the comment regarding future work (lines 390/391).

Reviewer 3 Report

It is necessary to improve the Figures 4, 5 and 6 in terms of better detecting the traces that derive from one of the two radars, the fused trace and the trace that originates from the AIS data.

Since the testing of the algorithm lasts for more than a year, I think it would be useful to show the robustness of the algorithm in conditions where there is turbulent sea, strong wind and abundant atmospheric precipitation.

Author Response

Dear reviewer,

Thank you for the very positive review.

Detailed answers to your comments are provided below.

It is necessary to improve the Figures 4, 5 and 6 in terms of better detecting the traces that derive from one of the two radars, the fused trace and the trace that originates from the AIS data.

Figures are edited in order to clarify details. This is done with digital zooming of the trace area so the traces are easily visible now.

Since the testing of the algorithm lasts for more than a year, I think it would be useful to show the robustness of the algorithm in conditions where there is turbulent sea, strong wind and abundant atmospheric precipitation.

Strong wind and rough sea impact on target detection are more question of CFAR algorithm than multi sensor (HFSWR) fusion algorithm, since false alarms created by high waves should be eliminated in CFAR stage. The paper regrading CFAR is currently being written.

Moreover since the system became operational we noticed some phenomenon which may be regarded as meteo tsunami, since there are no available tide gauges and weather buoys in the region we contacted tsunami experts to help us with data analyses obtained via available HFSWR network. This findings are currently under detailed research and we hope that they will be published soon.

Best regrards,
Dejan Nikolic

Round 2

Reviewer 2 Report

Reviewers' Suggestions and Comments:

The authors have addressed satisfactorily the comments from the previous review round. In the present form, the paper looks to be published in the journal. So I accept this version for publishing.

1. The authors should have also checked their notations and abbreviations before paper submission. EX: AIS,  UN [2] and / or EU, HFWR ( in the text line 91),

SAIS ….

Author Response

Dear reviewer,

The paper is extended accordingly to your remark.

Best regards,

Dejan Nikolic